# Immunohistochemical Evaluation of Periodontal Regeneration Using a Porous Collagen Scaffold

**DOI:** 10.3390/ijms222010915

**Published:** 2021-10-09

**Authors:** Jean-Claude Imber, Andrea Roccuzzo, Alexandra Stähli, Nikola Saulacic, James Deschner, Anton Sculean, Dieter Daniel Bosshardt

**Affiliations:** 1Department of Periodontology, School of Dental Medicine, University of Bern, 3010 Bern, Switzerland; andrea.roccuzzo@zmk.unibe.ch (A.R.); alexandra.staehli@zmk.unibe.ch (A.S.); anton.sculean@zmk.unibe.ch (A.S.); dieter.bosshardt@zmk.unibe.ch (D.D.B.); 2Robert K. Schenk Laboratory of Oral Histology, School of Dental Medicine, University of Bern, 3010 Bern, Switzerland; 3Department of Cranio-Maxillofacial Surgery, Inselspital, Bern University Hospital, Faculty of Medicine, University of Bern, 3010 Bern, Switzerland; nikola.saulacic@insel.ch; 4Department of Periodontology and Operative Dentistry, University Medical Centre of the Johannes Gutenberg University, 55131 Mainz, Germany; james.deschner@uni-mainz.de

**Keywords:** periodontal regeneration, intrabony defect, collagen scaffold, volume-stable collagen matrix, biomaterial, histology, immunohistochemistry

## Abstract

(1) Aim: To immunohistochemically evaluate the effect of a volume-stable collagen scaffold (VCMX) on periodontal regeneration. (2) Methods: In eight beagle dogs, acute two-wall intrabony defects were treated with open flap debridement either with VCMX (test) or without (control). After 12 weeks, eight defects out of four animals were processed for paraffin histology and immunohistochemistry. (3) Results: All defects (four test + four control) revealed periodontal regeneration with cementum and bone formation. VCMX remnants were integrated in bone, periodontal ligament (PDL), and cementum. No differences in immunohistochemical labeling patterns were observed between test and control sites. New bone and cementum were labeled for bone sialoprotein, while the regenerated PDL was labeled for periostin and collagen type 1. Cytokeratin-positive epithelial cell rests of Malassez were detected in 50% of the defects. The regenerated PDL demonstrated a larger blood vessel area at the test (14.48% ± 3.52%) than at control sites (8.04% ± 1.85%, *p* = 0.0007). The number of blood vessels was higher in the regenerated PDL (test + control) compared to the pristine one (*p* = 0.012). The cell proliferative index was not statistically significantly different in pristine and regenerated PDL. (4) Conclusions: The data suggest a positive effect of VCMX on angiogenesis and an equally high cell turnover in the regenerated and pristine PDL. This VCMX supported periodontal regeneration in intrabony defects.

## 1. Introduction

Over the last decades, various surgical techniques were introduced aiming to regenerate the periodontal tissues (i.e., formation of new cementum, periodontal ligament (PDL), bone, and junctional epithelium (JE)) after their loss due to periodontal disease [1,2,3,4,5,6,7,8,9,10,11]. Various biomaterials such as bone grafts/bone substitutes, resorbable and non-resorbable membranes, enamel matrix derivative, growth and differentiation factors, or different combinations thereof were used to achieve periodontal regeneration [1,2,3,4,6,7,8,12,13].

For a successful clinical outcome of periodontal regenerative therapy, the following factors are of utmost importance: (1) wound stability, which supports blood clot adhesion and maturation on a decontaminated root surface; (2) space provision to give the cells from the bone and PDL including their blood vessels the opportunity to grow into the defect area where they can regenerate the periodontium; and (3) uneventful healing without bacterial contamination [1,2,3,6,7,14]. Thus, novel biomaterials supporting periodontal regeneration should be designed to facilitate these critical factors.

Collagen-based scaffolds have shown to have a great potential in hard and soft tissue engineering due to their ability to provide an environment closely resembling the native extracellular matrix [15,16,17,18]. Different collagen-based biomaterials are available for dental use with a variety of clinical indications [19,20].

In preclinical and clinical studies, a volume-stable collagen scaffold (VCMX) has been shown to possess excellent biocompatibility, favorable soft connective tissue integration, and enhanced angiogenesis [15,21,22,23,24]. This biomaterial is mainly used for soft tissue augmentation procedures. Nevertheless, due to its structural configuration (e.g., high porosity, interconnectivity and cross-linking) [25], the material has the potential to serve as a scaffold for PDL cells and, consequently, may facilitate periodontal regeneration.

Therefore, the aim of this study was to immunohistochemically evaluate the effect of the VCMX scaffold on periodontal regeneration. An antibody against cytokeratins (CKs) was used to visualize both the apical growth of the regenerated JE and the epithelial rests of Malassez (ERM). Immunohistochemical detection of bone sialoprotein (BSP) was chosen because this non-collagenous protein is expressed in cementoblasts [26], regulates mineralization [27], and is essential for cementogenesis and periodontal function [28]. The reason to perform immunohistochemistry for cementum attachment protein (CAP) was based on the assumption that this protein may serve as a specific cementum marker [29,30] and may play an important role in cell recruitment and cementoblast differentiation [29,31,32]. Collagen type 1 (COL1) and periostin (PER) immunohistochemistry was applied since these two proteins are known to be highly expressed in the PDL, while PER also plays an essential role in ensuring periodontal tissue integrity [33,34,35]. Finally, an antibody against proliferating cells (anti-proliferative cell nuclear antigen antibody, PCNA) was used to visualize and quantitate the cell proliferative activity in the regenerated PDL.

## 2. Results

The healing was uneventful in all dogs without wound dehiscence or other major complications. All eight defects embedded in paraffin were available for descriptive analysis. The histologic analysis revealed periodontal regeneration evidenced by formation of cementum, PDL, and bone to a varying extent in both the test and control group (Figure 1). In the test group, remnants of the VCMX were observed in all four defects. These remnants were embedded in new gingival connective tissue, new bone, new PDL, and new cementum (Figure 1a,d,g). Root resorptions or other adverse events like inflammation that could be related to the use of the VCMX were not observed.

### 2.1. Immunohistochemistry

#### 2.1.1. Negative Controls

The negative controls for all the different immunohistochemical incubations did not show any positive labeling (not shown).

#### 2.1.2. Anti-Cytokeratin Antibody

Epithelial cells were heavily labeled with the anti-CK antibody (Figure 2a,b). The inspection of the sections revealed an absence of the formation of a long JE, irrespective of test or control group. The JE had similar anatomical features at test and control sites (e.g., tapering off in the apical direction, and absence of epithelial ridges). Interestingly, ERM were observed in the regenerated PDL of 50% of the test (Figure 3b) and 50% of the control sites (Figure 3c). If ERM was detected in the pristine PDL (Figure 3a), they were observed as well in the regenerated PDL (Figure 3b,c) and vice versa (Figure 3d–f).

#### 2.1.3. Anti-Bone Sialoprotein Antibody

BSP labeling was observed in both cementum and bone, irrespective of newly formed or pristine hard tissues (Figure 4a–d). In bone, BSP labeling was highest at the periphery of osteocyte lacunae, over reversal and resting lines, and at the bone-PDL interface, whereas the immunostaining of the bone matrix was moderate (Figure 4). For cementum, the highest BSP labeling was found at the interface between the treated root surface and new cementum (Figure 4b,c), whereas the cementum matrix labeling was moderate (Figure 4a–d). In some defects, BSP-labeled cementum was observed right to the apical end of the JE (Figure 2c,d). Additionally, weak BSP immunostaining was seen in the soft connective tissues of the gingiva and PDL with weaker staining in pristine compared to the newly formed tissue (Figure 4a–d).

#### 2.1.4. Anti-Cementum Attachment Protein Antibody

Throughout the entire pristine PDL, reticular CAP labeling was observed which was associated with collagen (Figure 4e). While a distinct CAP labeled band was detected at the cementum-PDL interface, such a band did not exist at the bone-PDL interface (Figure 4e).

In the regenerated PDL at both test (Figure 4f) and control (Figure 4g) sites, less CAP labeling was found in the PDL compared to the pristine PDL (Figure 4e). Regarding the hard tissue-PDL interfaces, a CAP labeled band was detected at the cementum and bone surfaces, whereby the band immunostained for CAP was more continuous and thicker at the bone surface (Figure 4f,g). Furthermore, at regenerated sites (Figure 4f,g), more Sharpey’s fibers with CAP+ immunostaining were detected, compared to the pristine PDL (Figure 4e).

#### 2.1.5. Anti-Collagen Type 1 Antibody

COL1 labeling was found throughout the entire width of both the regenerated and pristine PDL (Figure 5a–c). The pristine PDL showed more immunostaining for COL1 on the bone side, compared to the cementum side (Figure 5a), whereas in the regenerated PDL the distribution of the labeling was more homogenous (Figure 5b,c) with a more reticular pattern at the test sites. The superficial layer of cementum and parts of the superficial bone surface demonstrated the highest labeling for COL1 (Figure 5a–c).

#### 2.1.6. Anti-Periostin Antibody

Immunostaining for PER was observed throughout the whole width of the PDL (Figure 5d–f) and in the periosteum at the external surface of the alveolar process used as internal control (not shown). The most homogenous distribution of PER was observed in the pristine PDL (Figure 5d), followed by the PDL at control sites (Figure 5f). The least even distribution of PER labeling was found in the PDL at the test sites (Figure 5e) where the PER immunostaining demonstrated a reticular pattern with voids devoid of PER labeling. From the bone crest in a coronal direction, the PER labeling was continuously decreasing and resulted in very weak labeling of the gingival soft connective tissue adjacent to the JE (not shown).

#### 2.1.7. Anti-Proliferative Cell Nuclear Antigen Antibody

PCNA+ cells were seen in large numbers in the PDL (Figure 6b–d) and in the basal cell layer of the epithelium used as an internal control (Figure 6a). In most of the samples, a higher number of PCNA+ cells were detected on the cementum side compared to the bone side (Figure 6b–d). Cementoblasts in direct contact with the newly formed cementum were often positive for PCNA.

### 2.2. Quantitative Analysis of Blood Vessels and Proliferating Cells

The data for blood vessel area/number and PI are demonstrated in Table 1. There was no difference in the blood vessel area between pristine and regenerated (test and control sites combined) PDL (11.66% versus 10.38%, *p* = 0.224; Figure 7a). Our main finding was that in the regenerated PDL, the test sites showed a significantly higher blood vessel area per mm^2^ PDL compared to the control sites (14.48% versus 8.04%, *p* = 0.0007; Figure 7b). Concerning the number of blood vessels/mm^2^ PDL, significantly more blood vessels were counted in the regenerated PDL (test and control combined) than in the pristine PDL (109.09 versus 68.28, *p* = 0.0121; Figure 7c), whereas no statistically significant difference was found between test and control sites (94.26 versus 128.17, *p* = 0.1142; Figure 7d). The percentage of PCNA+ cells in relation to the total number of cells (proliferative index; PI) in the PDL is shown in Figure 7e,f. There was no difference in the PI between the regenerated (test and control sites combined) and the pristine PDL (57.09% versus 56.39%, *p* = 0.642; Figure 7e). Moreover, no statistically significant difference was found in the PI between the regenerated PDL at test and control sites (56.81% versus 57.45%, *p* = 0.68; Figure 7f).

## 3. Discussion

The present study has immunohistochemically investigated the healing characteristics of acute-type two-wall intrabony defects following regenerative periodontal surgery using a VCMX. So far, the VCMX has been tested and used for volume gain in oral soft connective tissues. To the best of our knowledge, this is the first immunohistochemical investigation where this novel biomaterial was tested for periodontal regeneration. In the present study, this collagen scaffold showed excellent biocompatibility as demonstrated by ingrowth of bone, cementum, and soft connective tissue and the absence of inflammatory and foreign body giant cells. Moreover, the data suggest a positive effect of VCMX on angiogenesis and demonstrate an equally high cell turnover rate in the regenerated compared to the pristine PDL.

CKs are proteins which provide mechanical support and are involved in a variety of additional functions in epithelial cells [36,37]. In humans, 20 different CKs isotypes have been identified [38]. The pan-CK antibody used in this study reacts with a high number of CKs (No. 1, 2, 3, 4, 5, 6, 7, 8 10, 13, 14, 15 16, 19). Since healthy, diseased and regenerated gingival epithelium (No. 1, 2, 5, 6, 10, 11, 13, 14, 16, 17) and ERM (No. 5, 19) express a great complexity of CKs [37,39,40], the used pan-CK antibody is ideal to detect epithelial cells in periodontal tissues. Despite the fact that the function of the ERM is not fully understood, there is increasing evidence that these cells are involved in PDL homeostasis, prevention of ankylosis and root resorption, maintaining PDL space, and cementum repair and regeneration [41,42,43]. In the present study, ERM were detected in 50% of the samples in both the pristine and regenerated PDL. Interestingly, previous studies failed to demonstrate the presence of ERM in newly formed PDL after regenerative therapy [40,44,45]. One potential explanation for this difference could be that the studies of Sculean et al. [40,45] were performed in monkeys and not in dogs. In a study by Araujo et al. [44], the same species was used as in the present study, but with a different periodontal defect model (i.e., furcation degree III). Therefore, the defect location could play a role in the repopulation of the newly formed PDL with ERM. Moreover, in the study of Araujo et al. (36), no immunohistochemical evaluation against epithelial cells was performed and therefore, detection of ERM could have been more difficult. However, the results of our analysis have to be interpreted with caution because of the low number of samples. Future investigations on the repopulation of ERM in the regenerated PDL require a higher number of samples.

BSP, a noncollagenous protein, is a well-known and established immunohistochemical marker for mineralized tissues like cementum and bone [46,47,48,49,50]. Cementoblasts express many important proteins for the mineralization process of cementum, including BSP [26,51]. BSP is involved in cellular and molecular events involved in cementogenesis [52]. A preclinical study with a BSP knock-out cementoblast cell line revealed a significant decrease of the mineralization capacity, pointing out the importance of BSP for proper cementogenesis [27]. Additionally, a BSP null mouse model revealed that without BSP the formation of acellular cementum is reduced and the deposited cementum appears hypomineralized [28]. Loss of BSP caused progressive disorientation of the PDL due to structural defects in the cementum-PDL interface, highlighting the significance of BSP for periodontal function [28]. Furthermore, an immunohistochemical study with healthy and periodontitis-affected human teeth has shown that BSP could not be detected on exposed cementum (absence of overlying PDL) of periodontally compromised teeth [53]. The authors concluded that the lack of BSP may influence the ability for regeneration and new connective tissue attachment onto previously diseased root surfaces [53].

BSP and CK labeling together allowed a precise determination of coronal growth of cementum and apical growth of JE (Figure 2). In some defects, cementum and JE met in a butt-joint fashion. Furthermore, some defects showed a thin layer of BSP-positive cementum layer reaching up to the apical end of the JE (Figure 2). This may indicate that cementum formation in the most coronal root portion was about to start after 12 weeks of healing. The detection of this tiny cementum layer was only possible with immunohistochemical staining. Furthermore, the present study shows that the interface between dentin and newly formed cementum, and the surface of newly formed bone were heavily labeled for BSP. This labeling pattern corresponds with results from other studies [54,55]. Other noncollagenous proteins like osteopontin were used in previous immunohistochemical studies on cementum and/or bone formation [56]. However, osteopontin is frequently found in other tissues than bone and cementum such as blood plasma, saliva, gingival crevicular fluid, kidney, and vascular tissues and is secreted by macrophages [56,57,58,59]. Thus, OPN is less specific for mineralized tissues than BSP.

Expression of CAP has been demonstrated in cementoblasts of different species (e.g., humans and bovines) [29]. CAP has been characterized as a collagenous attachment protein which plays a role in cell recruitment and differentiation during cementum formation [29,31,32]. Data have suggested that CAP is restricted to cementum and could therefore serve as a specific cementum marker [29,30]. There is no cross-reactivity of the CAP antibody known with antibodies to collagen types III, V, XII, XIV or other attachment proteins in the cementum (e.g., fibronectin, BSP, vitronectin) [32]. In our canine study, CAP immunohistochemistry was not specific for cementum. Interestingly, the PDL was more heavily labelled for CAP than cementum and bone. Some homologies of CAP with type I, X, and XII collagens are a possible explanation for this labeling pattern [32,60]. In addition, a previous study on periodontal regeneration in canines demonstrated the same CAP labeling of the PDL as shown in the present study [61]. The authors concluded that this PDL labeling indicated active cementum formation. Nevertheless, since our study showed a stronger immunostaining for CAP in the pristine PDL compared to the regenerated PDL, we attribute this difference to a denser collagen network in the pristine PDL and the reactivity of the CAP antibody to certain collagenous or collagen-associated proteins. COL1 and PER, both extracellular matrix proteins, were immunohistochemically detected in the PDL.

PER is known to be expressed in the periosteum and PDL, with an essential function for periodontal tissue integrity [33,34,35]. A preclinical study has shown severe periodontal defects after tooth eruption in a PER-null mouse model and a certain resolution of the periodontal defects after removal of the masticatory forces [62]. Moreover, an in vitro investigation on strained PDL cells revealed a highly elevated expression of PER compared to unstimulated cells [62]. Consequently, the expression of PER is highly dependent on occlusal loading of the teeth and very important for the functional integrity of the periodontal apparatus. In a study by Park et al., it was demonstrated that PER can act as a marker for matrix maturation and stability [63]. Therefore, our results may suggest that the regenerated PDL is still immature. Because PER is much less expressed in gingival connective tissue compared to the PDL [33], the PER labeling allowed better visualization of the transition between PDL and gingival connective tissue. No striking differences between COL1 and PER labeling were observed in the PDL. This is in line with the known molecular interaction between PER and COL1 and the regulatory function of PER during fibrillogenesis [64]. The observed reticular labeling pattern for COL1 and PER in the PDL of the test group may be attributed to the porous nature of the VCMX. Indeed, a recent study has shown the sequential invasion of mesenchymal cells and the deposition of a collagenous matrix in the VCMX pores [21].

The PDL is a highly vascularized connective tissue connecting teeth to surrounding bone and permitting teeth to withstand the high forces of mastication [65]. It consists of cells (e.g., fibroblasts, ERM, osteoblasts, cementoblasts), collagenous and noncollagenous matrix constituents, and blood vessels [65]. The blood vessel area in the PDL of bovine teeth was investigated by Bosshardt et al. [66]. They showed a great variability of blood vessel area and number depending on the location in the PDL. The blood vessel area ranged from 11.9% to 23.5% and the number of blood vessels from 26 to 112 per mm^2^. In a study in beagle dogs, others found a blood vessel area of 10–20% in the PDL depending on the location [67]. In humans, an electron microscopy study revealed a blood microvascular luminal volume of 9.52% and an abluminal volume of 12.91% [68]. Therefore, our findings with 11.66% of blood vessel area within the newly formed PDL and 10.38% in the pristine PDL are in the range of data from other studies. Moreover, our results demonstrate statistically significant more blood vessel area in the regenerated PDL of the test group compared to the control group. This could be because the collagenous scaffold of the VCMX was promoting angiogenesis and/or the regeneration process of the PDL was still ongoing. It was shown that certain physical parameters of biomaterials including pore size and interconnectivity of pores can facilitate angiogenesis [69]. Ingrowth of blood vessels into the VCMX pores was recently demonstrated in a study on soft tissue augmentation [15].

The PI revealed a high cell proliferation rate, irrespective of pristine or regenerated PDL. Several studies have shown that the PDL has an extremely high turnover rate, much higher than the gingiva, skin, and bone [65,70,71,72,73]. Taken together, our data suggest that the VCMX did not negatively influence cell proliferation, indicating that the regenerated PDL had a cell turnover that was as high as the normal physiological turnover in the PDL.

A variety of biomaterials such as bone substitutes, growth factors, and barrier membranes are used to achieve periodontal regeneration [7]. Periodontal regenerative/reconstructive surgeries in combination with some of these biomaterials resulted in superior clinical outcomes (e.g., pocket depth reduction, clinical attachment level gain) compared to open flap debridement alone [3]. New concepts to regenerate all periodontal tissues may include the use of stem cells, bio-printing, gene therapy, and scaffold technologies alone or in combinations [74]. Nowadays, stem cell transplantation is a promising field in medicine but difficult to transfer into daily practice [75]. Since it is well known that only cells originating from the PDL have the potential to rebuild the periodontal attachment apparatus consisting of bone, PDL, and cementum [76,77,78,79], biomaterials acting as a scaffold to support the invasion of the host’s own progenitor/stem cells appear to be a promising avenue for future research and clinical applications. Endogenous cell homing using appropriate biomaterials can be regarded as a more economic, effective, and safe method for treating patients [80]. Therefore, recent research on biomaterials promoting periodontal regeneration is focusing on cell-free scaffold technologies for endogenous cell recruitment [81,82,83]. Studies have shown that the structural configuration of a scaffold has crucial implications on tissue engineering [83,84,85]. Recently, it was shown that a pore size of 100 μm was found to be necessary to ensure an even distribution of PDL cells across a scaffold cross-section [83]. The VCMX used in the present study has an average pore diameter of 92 μm [15]. Thus, a pore size of around 100 μm may be ideal for PDL regeneration.

The interpretation and translation of results from animal research to the human situation is one of the major difficulties. The different anatomical and physiological environments and healing rates are part of this problem. Furthermore, surgically created acute-type defects have a defect configuration that does not reflect the true situation of human periodontitis where the presence of bacteria and inflammatory reactions come into play [86,87]. A spontaneous regeneration of a certain amount of periodontal tissues can be expected in acute-type defects [88,89]. However, the preclinical setting used in this study is well established for evaluating periodontal regeneration [86,87,90,91]. The healing pattern is comparable to that of chronic defects with a decontaminated root surface and the defect configuration and size can be much more standardized in this type of defect [86,87]. A shortcoming of the present study may be that only one healing period was investigated. However, before studying the effects of a new biomaterial on the sequential events in wound healing and regeneration, the outcome should be tested as a proof of principle. Future studies on periodontal regeneration with the VCMX may investigate earlier wound healing events. Furthermore, a combination of collagen scaffolds with growth/differentiation factors should be addressed in future studies. Finally, the last step should include clinical studies to evaluate clinical, radiological, and patient-related outcomes of this biomaterial used in intrabony and/or suprabony defects in humans.

## 4. Materials and Methods

This preclinical study was approved by the Ethics Committee of the Rof Codina Foundation, Lugo, Spain (02/16/LU-001). The Guidelines for Animal Research: Reporting In Vivo Experiments (ARRIVE) [92] have been included.

### 4.1. Animals & Surgical Procedure

Detailed information about the animal model and the surgical procedure can be found in a previous publication [93]. In brief, eight 18–24 months old beagle dogs with an intact dentition and a healthy periodontal status were used. All the surgical procedures were performed in the maxilla. The second and fourth premolars of both maxillary quadrants were extracted. After a healing period of 12 weeks, acute type 2-wall intrabony defects were surgically created distal to the first and third premolar of each dog and the root cementum was carefully removed. With a randomized assignment, the defects were either filled with a porous and volume stable collagen scaffold (VCMX, Geistlich Fibro-Gide^®^, Geistlich Pharma AG, Wolhusen, Switzerland) in the test group or were left empty in the control group. Subsequently, the flaps were closed to allow primary intention healing. The animals were euthanized 12 weeks after this procedure.

### 4.2. Histological Processing & Descriptive Analysis

After euthanization, 32 block biopsies were harvested and subsequently fixed in 10% formalin. In four out of eight animals, one test and one control site were randomly selected for paraffin histology. Consequently, eight defects (four test and four control sites) were decalcified in 10 % ethylenediaminetetraacetic acid and embedded in paraffin, whereas the remaining defects were processed to produce undecalcified ground sections. The histometric results have recently been published [93]. In the present study, the paraffin blocks were used for immunohistochemical evaluation. The paraffin blocks were cut in a mesiodistal plane and parallel to the long axis of the teeth using a microtome set at 8 μm. Tissue sections were stained with hematoxylin/eosin (H&E) and Masson’s trichrome. Immunohistochemistry was performed with antibodies against CKs [94] for epithelial cells, BSP and CAP for mineralized tissues, COL1 and PER for PDL, and proliferating cell nuclear antigen (PCNA) for proliferating cells.

For all processing procedures, micrographs were taken using a digital camera (AxioCam MRc; Carl Zeiss, Oberkochen, Germany) connected to a light microscope (Axio Imager M2; Carl Zeiss, Oberkochen, Germany).

### 4.3. Immunohistochemistry

Immunohistochemical evaluation was performed at pristine (non-treated sites) and regenerated periodontal tissues (treated sites) in both the test and control groups. For all antibodies used, a negative control where the primary antibody was omitted, was performed.

#### 4.3.1. Anti-Cytokeratin Antibody

Paraffin sections were selected and deparaffinized. Thereafter, the sections were stained with a pan anti-CK antibody, (Clones AE1/AE3, Dako M3515, Agilent Technologies, Santa Clara, CA, USA) with a dilution of 1:50. The antibody was applied for 1 h at room temperature. The Dako EnVisionTM + Dual Link System-HRP (DAB+) was used (Agilent Technologies, Santa Clara, CA, USA). Counterstaining was performed using Mayer’s hematoxylin solution (Merck, Darmstadt, Germany).

#### 4.3.2. Anti-Bone Sialoprotein Antibody

Paraffin sections were selected and deparaffinized. Non-specific binding was blocked using defatted milk for 30 min. Thereafter, the sections were stained with an anti-BSP antibody (LF-120, Kerafast, Boston, MA, USA), diluted at 1:100, for 1 h at room temperature. The Dako EnVisionTM + Dual Link System-HRP (DAB+) was used (Agilent Technologies, Santa Clara, CA, USA). Counterstaining was performed using Mayer’s hematoxylin solution (Merck, Darmstadt, Germany).

#### 4.3.3. Anti-Cementum Attachment Protein Antibody

Paraffin sections were selected and deparaffinized. Non-specific binding was blocked using 3% H_2_O_2_ for 10 min. Thereafter, the sections were stained with an anti-CAP antibody (3G9, sc-53947, Santa Cruz Biotechnology, Dallas, TX, USA) with a dilution of 1:50. The antibody was applied for 1 h at room temperature. The Dako EnVisionTM+Dual Link System-HRP (DAB+) was used (Agilent Technologies, Santa Clara, CA, USA). Counterstaining was performed using Mayer’s hematoxylin solution (Merck, Darmstadt, Germany).

#### 4.3.4. Collagen Type 1 Anti-Collagen Type 1 Antibody

Paraffin sections were selected and deparaffinized. Epitope retrieval was accomplished with heat (80–82 °C) in a citrate solution. Non-specific binding was blocked using defatted milk for 30 min. Thereafter, the sections were stained with an anti-COL1 antibody (ab6308, Abcam, Cambridge, UK) with a dilution of 1:100. The antibody was applied for 1 h at room temperature. The Dako EnVisionTM + Dual Link System-HRP (DAB+) was used (Agilent Technologies, Santa Clara, CA, USA). Counterstaining was performed using Mayer’s hematoxylin solution (Merck, Darmstadt, Germany).

#### 4.3.5. Anti-Periostin Antibody

Paraffin sections were selected and deparaffinized. Thereafter, the sections were stained with an anti-PER antibody (ab14041, Abcam, Cambridge, UK) with a dilution of 1:500 for 1 h at room temperature. The Dako EnVisionTM + Dual Link System-HRP (DAB+) was used (Agilent Technologies, Santa Clara, CA, USA). Counterstaining was performed using Mayer’s hematoxylin solution (Merck, Darmstadt, Germany).

#### 4.3.6. Anti-Proliferative Cell Nuclear Antigen Antibody

Paraffin sections were selected for epitope retrieval with heat (87 °C) in a citrate solution. Non-specific binding was blocked using defatted milk for 30 min. Thereafter, the sections were stained with an anti-PCNA antibody (Clone PC 10, Dako M 0879, Agilent Technologies, Santa Clara, CA, USA) with a dilution of 1:50. The antibody was applied for 90 min at room temperature. The Dako EnVisionTM + Dual Link System-HRP (DAB+) was used (Agilent Technologies, Santa Clara, CA, USA). Counterstaining was performed using Mayer’s hematoxylin solution (Merck, Darmstadt, Germany).

### 4.4. Quantitative Analysis of Proliferating Cells and Blood Vessels

A quantitative analysis was performed for proliferating cells and blood vessels in the PDL using the PCNA stained sections. Two sections per defect were evaluated with the use of software (Zen pro, Carl Zeiss, Oberkochen, Germany). The method for the analysis of blood vessel number and area fraction was adapted from Bosshardt and coworkers [66]. Briefly, the PDL on the distal (treated sites) and mesial (pristine/non-treated sites) aspects of the root in all PCNA stained sections was outlined. At treated sites, the region of interest (ROI) in the PDL was defined starting from the alveolar crest to the apical end of the apical notch, whereas the ROI of the pristine PDL started after the apical curvature of the root and ended at the coronal most portion of the PDL. With the use of the software, all visible blood vessels were filled with a red background in the ROI. Visual control of the highlighted blood vessels was performed, and manual corrections were made if necessary. Subsequently, the software calculated both the area (mm^2^) of the ROI and the total area (mm^2^) and number of the blood vessels. Hence, the blood vessel area fraction (% of blood vessel area/mm2 PDL) and the numerical density (number of blood vessels/mm^2^ PDL) were analyzed. Thereafter, the cell proliferative index (PI) was determined as published in previous research [95,96,97,98,99]. In brief, PCNA-positive and PCNA-negative cells were counted in the ROI and the PI (i.e., number of positive cells vs. total number of cells, in percentage) was calculated.

### 4.5. Statistical Analysis

Data analyses were performed using Prism v7 (GraphPad Software, La Jolla, CA, USA). Data are presented as medians and interquartile ranges unless stated otherwise. Statistical analysis was performed for area of blood vessels, number of blood vessels and PI. Outcomes were compared using Mann–Whitney U test due to the sample size and the not normally distributed samples.

## 5. Conclusions

The present results demonstrate that the VCMX: (1) had excellent biocompatibility as demonstrated by the ingrowth of bone, cementum, and soft connective tissue and absence of inflammation, (2) had a positive effect on angiogenesis, and (3) led to a PDL cell turnover that was as high as that of the pristine PDL.

## Figures and Tables

**Figure 1 ijms-22-10915-f001:**
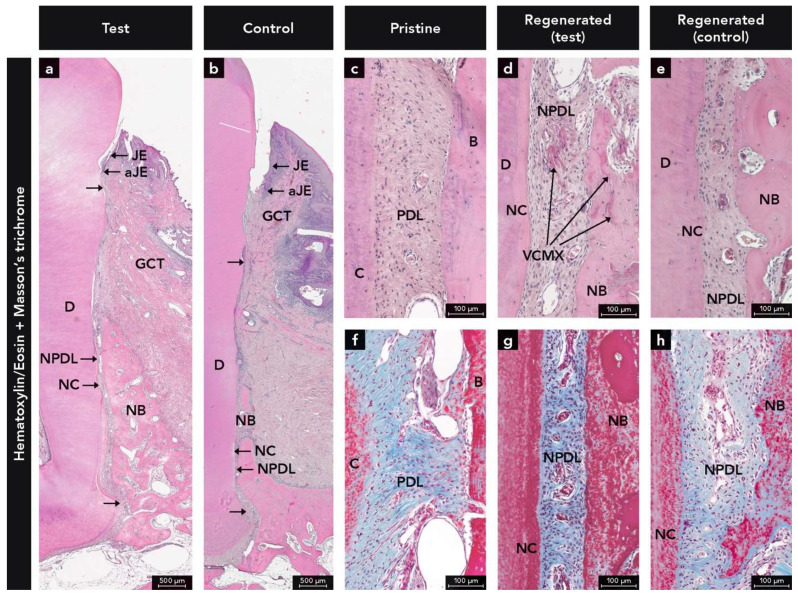
Micrographs illustrating periodontal regeneration (i.e., formation of a new junctional epithelium (JE), new cementum (NC), new periodontal ligament (NPDL), and new bone (NB)) at (**a**,**d**,**g**) test and (**b**,**e**,**h**) control sites; (**c**,**f**) Paraffin sections demonstrating the periodontal ligament (PDL), cementum (C), and bone (B) at non-treated/pristine sites; (**d**) Micrograph illustrating the integration of the volume stable collagen matrix (VCMX) into NPDL and NB; Paraffin sections staining: (**a**–**e**) hematoxylin & eosin and (**f**–**h**) Masson’s trichrome; aJE, apical end of the JE; D, dentin; GCT, gingival connective tissue.

**Figure 2 ijms-22-10915-f002:**
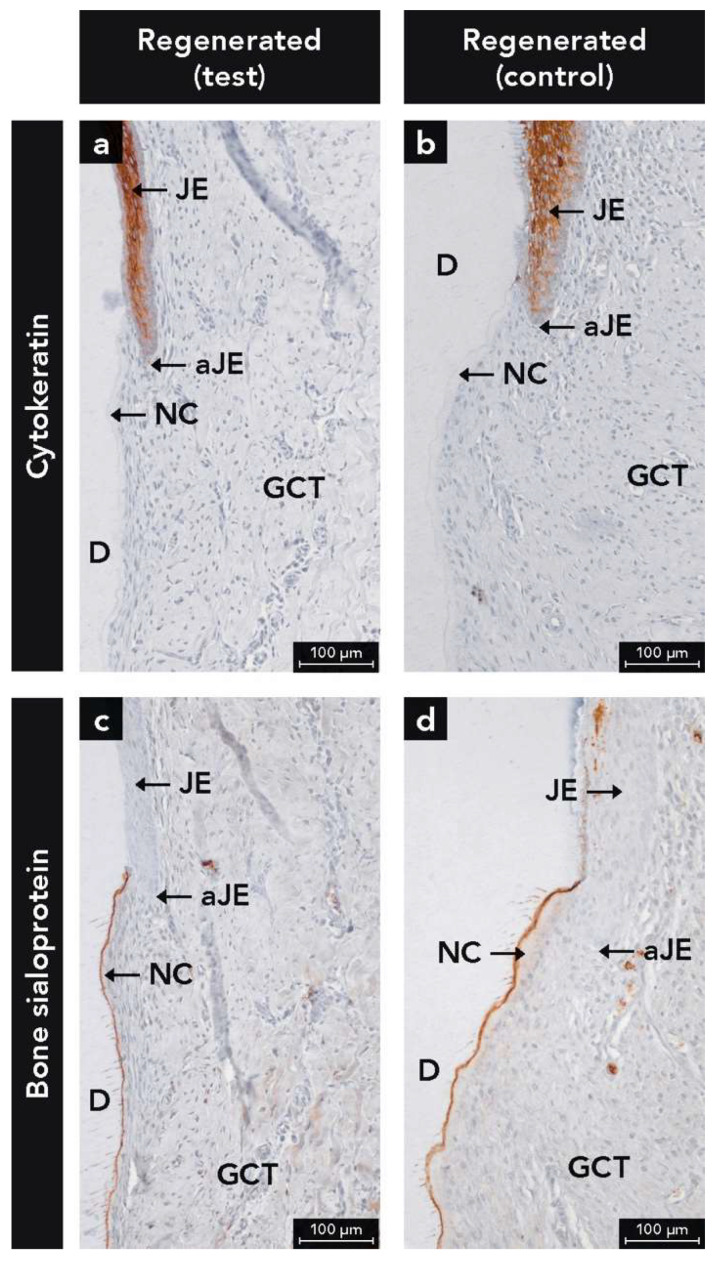
(**a**,**b**) Micrographs illustrating immunostained epithelial cells of the junctional epithelium (JE) with the use of a pan-cytokeratin antibody; (**c**,**d**) Micrographs showing immunostained new cementum (NC) with the use of an antibody against bone sialoprotein; Counterstaining: Mayer’s hematoxylin; aJE, apical end of the JE; D, dentin; GCT, gingival connective tissue.

**Figure 3 ijms-22-10915-f003:**
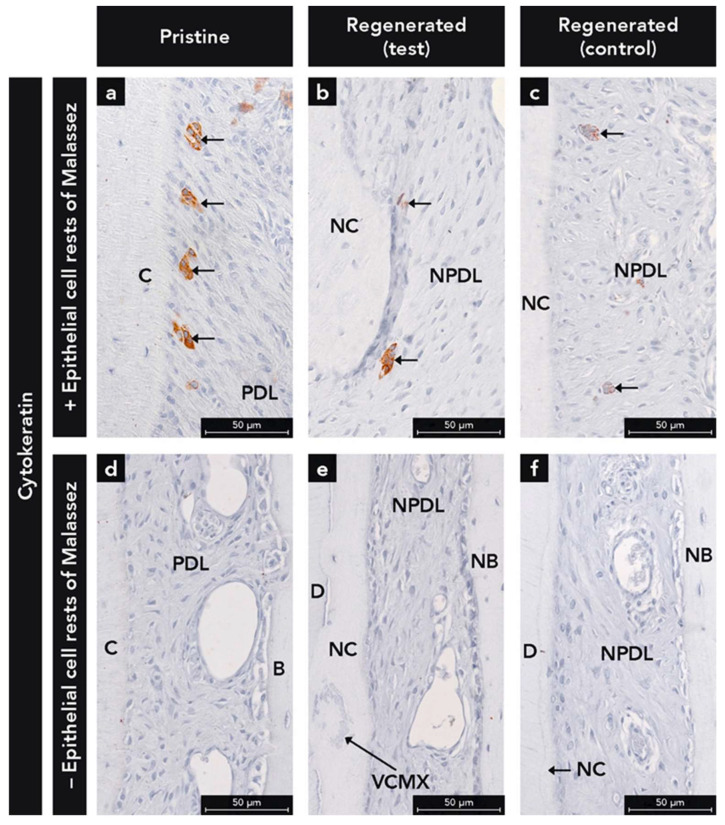
Micrographs demonstrating histological sections (**a**–**c**) with epithelial cell rests of Malassez (arrows) in periodontal ligament (PDL) and new periodontal ligament (NPDL) as demonstrated by immunostaining with the use of a pan-cytokeratin antibody and (**d**–**f**) without epithelial cell rests of Malassez; (**e**) Integration of the volume stable collagen matrix (VCMX) in new cementum (NC); Counterstaining: Mayer’s hematoxylin; C, cementum; D, dentin; NB, new bone.

**Figure 4 ijms-22-10915-f004:**
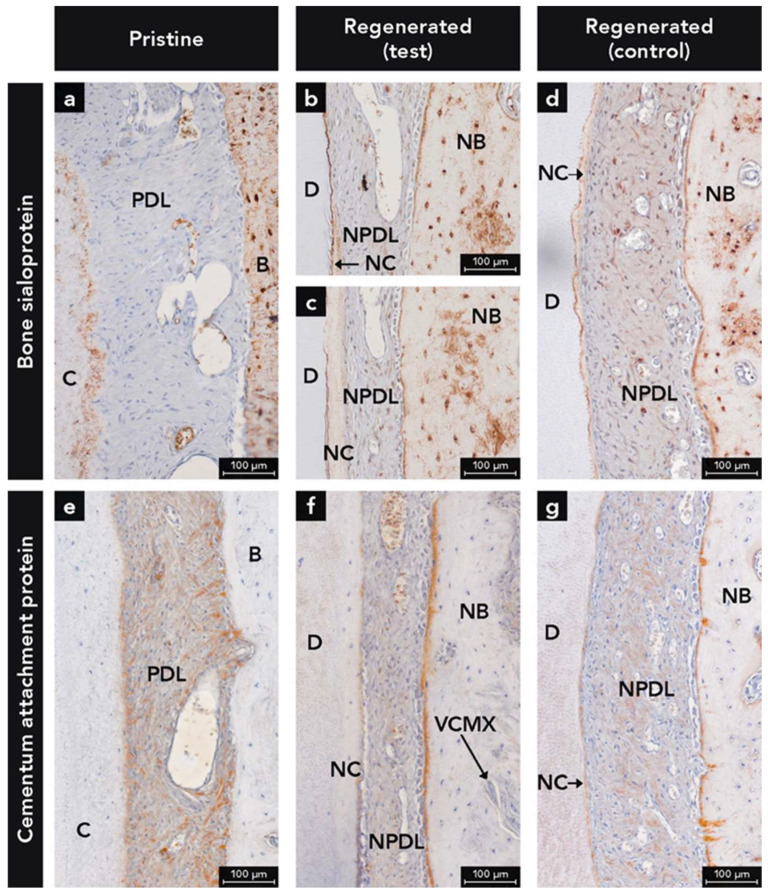
Micrographs demonstrating immunohistochemical staining for (**a**–**d**) bone sialoprotein antibody and (**e**–**g**) cementum attachment protein; (**f**) Integration of residues of the volume stable collagen matrix (VCMX) in new bone (NB); Counterstaining: Mayer’s hematoxylin solution; B, bone; C, cementum; D, dentin; NC, new cementum; NPDL, new periodontal ligament; PDL, periodontal ligament.

**Figure 5 ijms-22-10915-f005:**
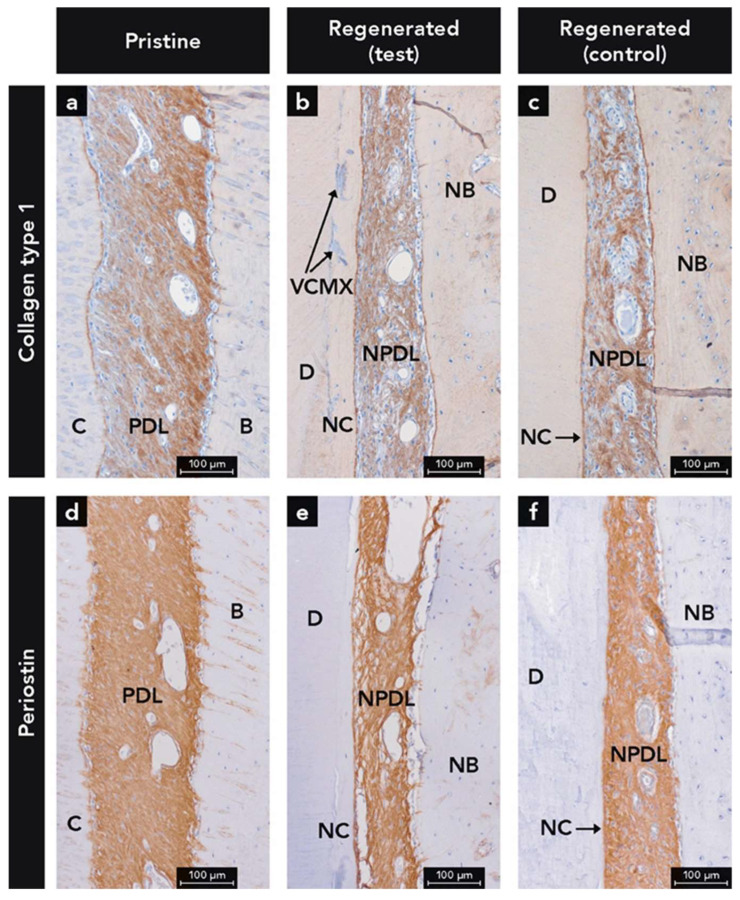
Micrographs demonstrating immunohistochemical staining for (**a**–**c**) collagen type 1 and (**d**–**f**) periostin; (**b**) Integration of residues of the volume stable collagen matrix (VCMX) in new cementum (NC); Counterstaining: Mayer’s hematoxylin; B, bone; C, cementum; D, dentin; NB, new bone; NPDL, new periodontal ligament; PDL, periodontal ligament.

**Figure 6 ijms-22-10915-f006:**
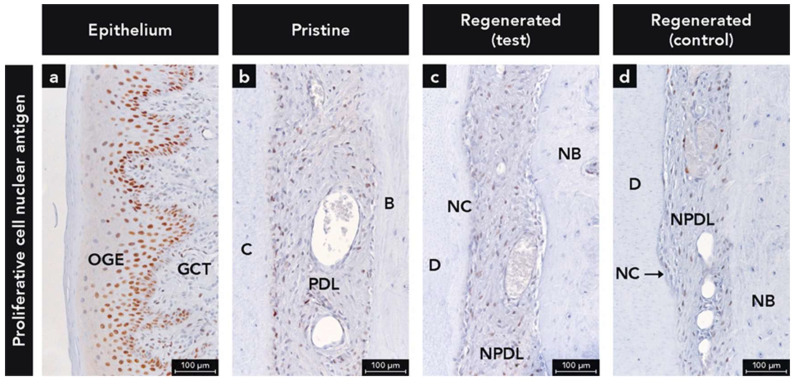
Immunohistochemical staining for proliferative cell nuclear antigen (PCNA); (**a**) Micrograph demonstrating PCNA+ cells in the oral gingival epithelium (OGE); (**b**–**d**) Micrographs showing PCNA+ cells in the pristine and regenerated periodontal ligament (PDL); Counterstaining: Mayer’s hematoxylin; B, bone; C, cementum; D, dentin; GCT, gingival connective tissue; NB, new bone; NC, new cementum.

**Figure 7 ijms-22-10915-f007:**
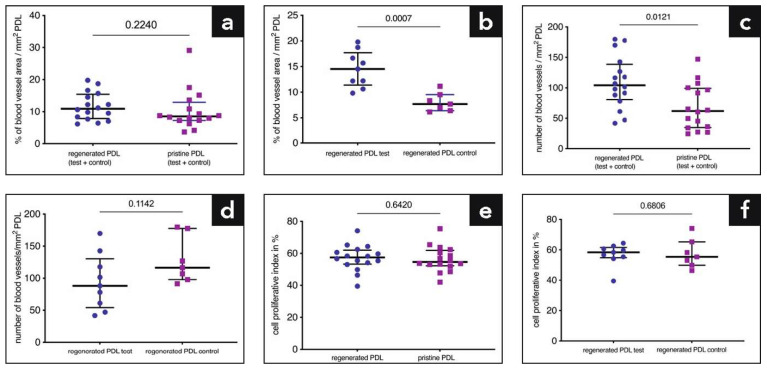
(**a**,**b**) Graphs illustrating the percentage of blood vessel area per mm^2^ periodontal ligament (PDL); (**c**,**d**) Graphs illustrating the number of blood vessels per mm^2^ PDL; (**e**,**f**) Graphs demonstrating the cell proliferative index (i.e., number of proliferative cell nuclear antigen positive cells vs. total number of cells in percentage). The bars depict the median and the whiskers the interquartile range. Each dot represents a quantitatively analyzed PCNA stained section.

**Table 1 ijms-22-10915-t001:** Data for blood vessel area/number and cell proliferative index (PI).

	% of Blood Vessel Area/mm^2^		Number of Blood Vessels/mm^2^		Proliferative Index PI	
	Mean, SD	Median	Mean, SD	Median	Mean, SD	Median
Groups						
Regenerated test ^1^	14.48 ± 3.52	10.91	94.26 ± 43.40	88.14	56.81 ± 7.26	58.36
Regenerated control ^2^	8.05 ± 1.85	7.70	128.17 ± 36.45	116.50	57.45 ± 9.50	55.38
Regenerated total t+c ^3^	11.66 ± 4.33	10.91	109.09 ± 42.87	104.30	57.09 ± 8.02	57.44
Pristine test ^4^	11.56 ± 7.37	8.29	63.01 ± 35.22	49.56	55.25 ± 6.93	53.79
Pristine control ^5^	8.86 ± 4.34	8.73	78.21 ± 39.49	78.39	57.85 ± 9.50	56.80
Pristine total t+c ^6^	10.38 ± 6.20	8.51	68.28 ± 37.30	61.89	56.39 ± 7.97	56.39
*p* value 1 vs. 2	0.007		0.114		0.680	
*p* value 3 vs. 6	0.224		0.012		0.642	
*p* value 4 vs. 5	0.680		0.524		0.757	

t, test; c, control; ^1^ Regenerated test sites; ^2^ Regenerated control sites; ^3^ Regenerated test and control sites; ^4^ Pristine test sites; ^5^ Pristine control sites; ^6^ Pristine test and control sites.

## Data Availability

The data presented in this study are available on request from the corresponding author.

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
