# Peer review of "Immunohistochemical Evaluation of Periodontal Regeneration Using a Porous Collagen Scaffold"

_ijms, 2021, doi:10.3390/ijms222010915_

Round 1

Reviewer 1 Report

Major revision

Abstract:

The number of animals mentioned in the abstract is four while in the M&M the number is 8. The authors should clarify this point.

The authors state that more cementum and bone formed in the test group. These results are based on a previous study and should be removed from the abstract since, in the present study, only one histological section is presented to the reader with no statistical data.

M&M

4.2. Histological Processing & Descriptive Analysis

The authors should specify from how many animals they prepared the 8 paraffin blocks (2 or more).

4.3. Immunohistochemistry

The authors address the untreated control group (negative control) as control regenerated periodontal tissue. Since no regenerative treatment was performed to this group the name should be changed throughout the text and figures. The current terminology could be confusing to the reader.

  1. Conclusions

Line 429-430: " The present results demonstrate that the VCMX: (1) supported periodontal regeneration in intrabony defects in canines". As mentioned before, in the present study no measurements of bone/cementum formation have been performed and the authors need to rephrase this sentence.   

Results

Line 67-68: " The greater vertical extension of newly formed cementum and bone in the test group has been proven histometrically as reported recently [27]" Again, these findings are based on a previous study and the current study does not support them since no measurements or statistics were performed. The sentence should be removed.

Minor revision

Line 429-430: " The present results demonstrate that the VCMX: (1) supported periodontal regeneration in intrabony defects in canines". The word canines could be mistakenly interpreted by the reader as the type of teeth instead of beagle dogs.

Line 281: " The PI reveled a high cell proliferation rate" change to revealed

Line 281: change "ore" to or

Reviewer 2 Report

Dear authors, 

thank you for this interesting and nicely conducted study. It gives new valuable information about the potential use of a collagenous scaffold in periodontal regenerative procedures. 

In the discussion section there are several suggestions for BSP and Periostin part. Please add these references and information to the text. It will give a better perspective why the expression of these proteins are important for proper periodontal regeneration. 

Firstly, BSP  and many other important mineralization proteins are expressed in cementoblasts (Bozic D, Grgurevic L, Erjavec I, Brkljacic J, Orlic I, Razdorov G, Grgurevic I, Vukicevic S, Plancak D. The proteome and gene expression profile of cementoblastic cells treated by bone morphogenetic protein-7 in vitro. J Clin Periodontol. 2012 Jan;39(1):80-90. doi: 10.1111/j.1600-051X.2011.01794.x.) please add this information. Furthermore, it seems that there is defective mineralization by Bsp knock-out cementoblasts (Ao M, Chavez MB, Chu EY, Hemstreet KC, Yin Y, Yadav MC, Millán JL, Fisher LW, Goldberg HA, Somerman MJ, Foster BL. Overlapping functions of bone sialoprotein and pyrophosphate regulators in directing cementogenesis. Bone. 2017 Dec;105:134-147. doi: 10.1016/j.bone.2017.08.027.) further indicating its importance in proper cementogenesis and seems to be essential for periodontal function (Foster BL, Soenjaya Y, Nociti FH Jr, Holm E, Zerfas PM, Wimer HF, Holdsworth DW, Aubin JE, Hunter GK, Goldberg HA, Somerman MJ. Deficiency in acellular cementum and periodontal attachment in bsp null mice. J Dent Res. 2013 Feb;92(2):166-72. doi: 10.1177/0022034512469026. )

Furthermore, human immunohistochemistry has shown that bone sialoprotein was not detected in the exposed cementum (absence of overlying periodontal ligament) of diseased teeth, and the authors suggested that this may influence the ability of tissue to achieve regeneration. .( Lao M, Marino V, Bartold PM. Immunohistochemical study of bone sialoprotein and osteopontin in healthy and diseased root surfaces. J Periodontol. 2006 Oct;77(10):1665-73. doi: 10.1902/jop.2006.060087.)

Please add these references and information in this part. 

In the periostin part of the discussion please add the following reference (Rios HF, Ma D, Xie Y, Giannobile WV, Bonewald LF, Conway SJ, Feng JQ. Periostin is essential for the integrity and function of the periodontal ligament during occlusal loading in mice. J Periodontol. 2008 Aug;79(8):1480-90. doi: 10.1902/jop.2008.070624.)that deals with its important role in proper periodontium formation and discus it. 

In the end i congratulate the authors for a nice and excellent study. 

Reviewer 3 Report

EXCELLENT STUDY, very well conducted with high quality histological and immunohistochemical micrographs

the results are well presented and the limitations of this kind of study are well exposed in the discussion

However,  i don't understand why the material and method apperas as the fourth part of the study

Reviewer 4 Report

This study addresses an important topic in the periodontal regeneration. I have some specific comments, mostly related to the Methods section, below. 1.How the sample size was defined? 2. Have the authors considered to quantiatively evaluate the areas stained by DAB across groups (i.e., using Image J and some specific pulg-ins, such as the colour deconvolution plug-in) 3. Regarding the Quantitative Analysis of Proliferating Cells and Blood Vessels: -Is this a validated method? If yes, please add the appropriate reference. -How was the visual control of the highlighted blood vessels performed? How many examiners were involved in this process? 4. Please sate the limitations and strengths of this study at the end of the discussion section.

Round 2

Reviewer 1 Report

The authors have addressed all my concerns and the manuscript is significantly improved. I recommend the manuscript for publication.

Reviewer 4 Report

The answers presented by the authors were enough to clarify my question. I recommend the article for publication.